# Chaos enhancement in large-spin chains

Yael Lebel,[1] Lea F. Santos,[2] and Yevgeny Bar Lev[1]

[1]*Department of Physics, Ben-Gurion University of the Negev, Beer-Sheva 84105, Israel*
[2]*Department of Physics, Yeshiva University, New York, New York 10016, USA*

We study the chaotic properties of a large-spin XXZ chain with onsite disorder and a small number of excitations above the fully polarized state. We show that while the classical limit, which is reached for large spins, is chaotic, enlarging the spin suppresses quantum chaos features. We attribute this suppression to the occurrence of large and slow clusters of onsite excitations, and propose a way to facilitate their fragmentation by introducing additional decay channels. We numerically verify that the introduction of such relaxation channels restores chaoticity for large spins, so that only three excitations are required to achieve strong level repulsion and ergodic eigenstates.

## I. INTRODUCTION

Partially motivated by the Bohigas-Giannoni-Schmit conjecture [1, 2], quantum chaos was extensively studied in the 1980's and 1990's [3–5]. In the last ten years, the subject has seen a resurgence of interest due to its strong connection with several questions currently studied experimentally and theoretically, that include the issue of thermalization in isolated many-body quantum systems [6–8], the problem of heating in driven systems [9, 10], the difficulty to achieve many-body localization [11–14], and the fast scrambling of quantum information [15–19]. For systems with clear classical or semi-classical limits, quantum chaos refers to signatures found in the quantum domain, such as level statistics as in full random matrices [20], that indicate whether the classical system is chaotic in the sense of positive Lyapunov exponent and mixing. While this correspondence holds well for some systems with a small number of degrees of freedom, such as Sinai's billiard [1, 2], it has recently been shown to be violated in triangular billiards [21] and quantum triangle maps [22]. As one moves to systems with many interacting particles, this issue gets even more complicated, since the classical limit is not always straightforward [23].

In this work, we investigate a one-dimensional system of many interacting spins described by the Heisenberg XXZ model with nearest-neighbor couplings and onsite disorder. We employ the term "quantum chaos" as a synonym for level statistics as in random matrices. For the $z$-direction magnetization close to zero, this model is chaotic for spin-$1/2$ [24], spin 1 [25, 26], and larger spins [27, 28]. For spin-$1/2$, the model has also been shown to demonstrate chaotic traits for as little as 3 or 4 excitations above a fully polarized state of spins [29, 30] and even for a chain of only 3 spins-$1/2$ [31]. Here, we extend this analysis and examine the case of large-spin chains with a low number of excitations. In the semi-classical limit of a continuous spin, we verify that the system has a positive Lyapunov exponent. This might suggest that increasing the spin size would require even fewer excitations than in the case of spin-$1/2$ to reach the quantum chaotic regime. However, rather counterintuitively, the opposite takes place, larger spin takes us away from quantum chaos which we attribute to the emergence of clusters of onsite excitations. Therefore to achieve quantum chaos in the zero-density excitations limit, we need to modify the Hamiltonian by adding terms that ensures the fragmentation of those clusters. By doing so, quantum chaos can finally be reached with only 3 excitations.

## II. THE MODEL

We consider the large-spin version of the XXZ model with onsite disorder and open boundaries described by the following Hamiltonian

$$\hat{H} = \frac{J_{xy}}{s(s+1)} \sum_{k=1}^{L-1} \left( \hat{S}_x^k \hat{S}_x^{k+1} + \hat{S}_y^k \hat{S}_y^{k+1} \right) \tag{1}$$
$$+ \frac{J_z}{s(s+1)} \sum_{k=1}^{L-1} \hat{S}_z^k \hat{S}_z^{k+1} + \frac{1}{\sqrt{s(s+1)}} \sum_{i=1}^{L} h_k \hat{S}_z^k,$$

where $\hat{S}_\alpha^k$, with $\alpha = x, y, z$, stands for spin-$s$ operators acting on a lattice site $k$ with eigenvalues $S_\alpha^k \in [-s, s]$. The parameter $J_{xy}$ corresponds to the coupling strength in the $xy$ plane and $J_z$ stands for the strength of the interaction along the $z$-axis. To stay away from the isotropic point, $J_{xy} = J_z$, we choose $J_{xy} = 1$ and $J_z = 0.55$. The onsite disorder, where $h_k$ is independent and uniformly distributed random numbers in the interval $[-W, W]$, is introduced to break spatial symmetries. We use a weak amplitude, $W = 0.5$, to avoid possible localization effects. The model conserves the total $z$−magnetization $\sum_k \hat{S}_k^z$.

Since we are interested in the large spin limit of the model, the couplings and disorder parameters are normalized, such that all terms in the Hamiltonian have the same magnitude. For spin-$1/2$ and $W \sim J_{xy} \sim J_z$, model (1) is known to be chaotic [24, 32, 33].

To study the classical limit, it is convenient to normalize the spin operators, $\hat{\mathcal{S}}_\alpha^k = \hat{S}_\alpha^k / \sqrt{s(s+1)}$, which amounts to fixing the largest eigenvalue of the $\hat{\mathcal{S}}^2 =$

$\sum_k \left(\hat{\mathcal{S}}_\alpha^k\right)^2$ operator to 1. Using the normalized spin operators, the quantum Hamiltonian is given by

$$\hat{H} = J_{xy} \sum_{k=1}^{L-1} \left(\hat{\mathcal{S}}_x^k \hat{\mathcal{S}}_x^{k+1} + \hat{\mathcal{S}}_y^k \hat{\mathcal{S}}_y^{k+1}\right) \tag{2}$$
$$+ J_z \sum_{k=1}^{L-1} \hat{\mathcal{S}}_z^k \hat{\mathcal{S}}_z^{k+1} + \sum_{i=1}^{L} h_k \hat{\mathcal{S}}_z^k,$$

where the commutation relations of the *normalized* spins follow directly from the standard commutation relations of the spin,

$$\left[\hat{\mathcal{S}}_\alpha^k, \hat{\mathcal{S}}_\beta^k\right] = i \frac{1}{\sqrt{s(s+1)}} \epsilon_{\alpha\beta\gamma} \hat{\mathcal{S}}_\gamma^k, \tag{3}$$

where $\epsilon_{\alpha\beta\gamma}$ is the Levi-Civita symbol. In the $s \to \infty$ limit the normalized spins commute, which corresponds to the classical limit. We can then replace the operators $\hat{\mathcal{S}}_\alpha^k$ by real numbers $s_\alpha^k$, and obtain the classical version of the disordered XXZ model,

$$H_{cl} = J_{xy} \sum_{k=1}^{L-1} \left(s_x^k s_x^{k+1} + s_y^k s_y^{k+1}\right) \tag{4}$$
$$+ J_z \sum_{k=1}^{L-1} s_z^k s_z^{k+1} + \sum_{i=1}^{L} h_k s_z^k,$$

which represents classical interacting rotators $\vec{s}_i = \left(s_x^i, s_y^i, s_z^i\right)$ on a unit sphere. The classical system also conserves the total magnetization. We start by studying the chaotic properties of the model in the classical limit.

## III.   CLASSICAL CHAOS

To examine the chaotic properties of the classical Hamiltonian $H_{cl}$ in Eq. (4) , we examine the Lyapunov exponents starting from all the rotators pointing down in the $z$-direction, which corresponds to the lowest magnetization limit. The equations of motion of the rotators are obtained using Eq. (4) and Poisson brackets,

$$\frac{ds_\alpha^k}{dt} = \left\{H_{cl}, s_\alpha^k\right\}, \qquad \left\{s_\alpha^k, s_\beta^k\right\} = \delta_{kl} \epsilon_{\alpha\beta\gamma} s_\gamma^k, \tag{5}$$

which gives

$$\frac{ds_x^k}{dt} = -J_{xy}\left(s_y^{k+1} + s_y^{k-1}\right) s_z^k + \left[J_z\left(s_z^{k+1} + s_z^{k-1}\right) + h_k\right] s_y^k$$
$$\frac{ds_y^k}{dt} = J_{xy}\left(s_x^{k-1} + s_x^{k+1}\right) s_z^k - \left[J_z\left(s_z^{k+1} + s_z^{k-1}\right) + h_k\right] s_x^k$$
$$\frac{ds_z^k}{dt} = -J_{xy}\left[\left(s_x^{k-1} + s_x^{k+1}\right) s_y^k - \left(s_y^{k-1} + s_y^{k+1}\right) s_x^k\right], \tag{6}$$

and can be compactly written as nonlinear Bloch equations,

$$\frac{d\vec{s}_k}{dt} = -\vec{b}_k \times \vec{s}_k, \tag{7}$$

with an effective magnetic field,

$$\vec{b}_k = \vec{h}_k + \vec{J} \cdot \left(\vec{s}_{k+1} + \vec{s}_{k-1}\right), \tag{8}$$

where we defined the coupling vector $\vec{J} = (J_{xy}, J_{xy}, J_z)$. Since the equations conserve $s_k^2 = \vec{s}_k \cdot \vec{s}_k = 1$ of each rotator separately, the equation of one of the components of $\vec{s}$ is redundant and it is advantageous to use a numerical integration scheme which conserves all $s_k^2$ explicitly. One way to do this is to parametrize the orientation of the rotators on the unit sphere as $\vec{s}_k = (\sin\theta_k \cos\phi_k, \sin\theta_k \sin\phi_k, \cos\theta_k)$. This yields the following equations of motion for the angles,

$$\frac{d\phi_k}{dt} = J_{xy} \sin\theta_{k-1} \cot\theta_k \left(\cos\phi_{k-1}\cos\phi_k + \sin\phi_{k-1}\sin\phi_k\right)$$
$$+ J_{xy}\sin\theta_{k+1}\cot\theta_k\left(\cos\phi_{k+1}\cos\phi_k + \sin\phi_{k+1}\sin\phi_k\right)$$
$$- J_z\left(\cos\theta_{k+1} + \cos\theta_{k-1}\right) - h_k, \tag{9}$$
$$\frac{d\theta_k}{dt} = J_{xy}\sin\theta_{k+1}\left(\cos\phi_{k+1}\sin\phi_k - \sin\phi_{k+1}\cos\phi_k\right)$$
$$+ J_{xy}\sin\theta_{k-1}\left(\cos\phi_{k-1}\sin\phi_k - \sin\phi_{k-1}\cos\phi_k\right).$$

To calculate the Lyapunov exponents of this system we initialize all of the rotors with a slight deflection from the $z$ axis, $\delta\theta = \pi - \theta$, which corresponds to the low magnetization setting considered in this work. We set the angles $\phi_i$ in the $xy$ plane to be randomized, and integrate the equations of motion. The maximal Lyapunov exponent is calculated with the algorithm proposed in Ref. [34], which is based on finding the rate of growth of the fastest growing tangent space vector. The initial conditions in the azimuthal direction and the on-site disorder are changed randomly between realizations and the maximal exponents are then averaged. We use a chain of 50 spins to calculate the Lyapunov exponents and integrate the dynamics using the LSODA method described in Ref. [35] with time steps chosen optimally by the algorithm.

In Fig. 1 we present the results of the calculation for different deviations $\delta\theta$. It can be seen that even for low values of $\delta\theta$, which is equivalent to a low number of excitations in the quantum limit, the maximal Lyapunov exponent is positive, indicating that the classical limit is chaotic.

## IV.   QUANTUM CHAOS

After establishing that the classical limit of our model in Eq. (4) is chaotic at low magnetization, we proceed to examine whether the quantum Hamiltonian in Eq. (1) exhibits properties associated with quantum chaos. These

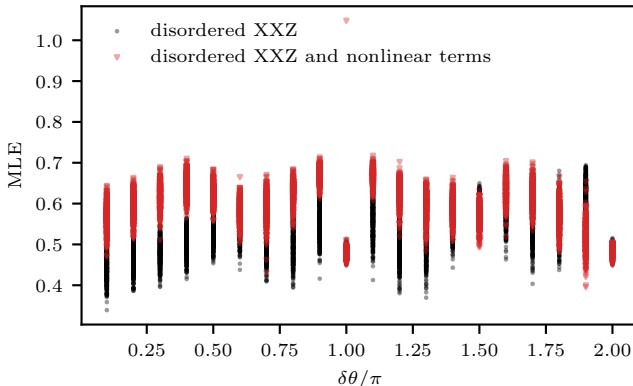

Figure 1. Maximal Lyapunov exponent (MLE) as a function of the deviation $\delta\theta$ (measured in radians) from the initial condition with all the spins pointing in the $z$-direction. Each point represents the average over different realizations for random values of $\phi_i$ and for the on-site disorder. The black circles correspond to the maximal Lyapunov exponent of the model in Eq. (4), while the red triangles correspond to the maximal Lyapunov exponent of the model in Eq. (13) with additional nonlinear terms. A chain of $L = 50$ is considered for these simulations.

properties include correlated eigenvalues, as in random matrix theory, and eigenstates that away from the edges of the spectrum are close to the eigenstates of full random matrices, that is, their components are nearly independent real random numbers from a Gaussian distribution satisfying the normalization condition [6, 7]. The onset of these almost random vectors in many-body quantum systems results in normal distributions of the off-diagonal elements of local observables [36–42], which is one of the features of the eigenstate thermalization hypothesis (ETH) [26, 43–51].

Our analysis of quantum chaos focuses on level statistics and the off-diagonal ETH. It is done for the subspace with total $z$-magnetization equal to $-sL + N$, where $N$ is a fixed number of excitations.

To study the level statistics, we use the so called $r$-metric [52–54],

$$r_\alpha = \min\left(\frac{\delta_\alpha}{\delta_{\alpha-1}}, \frac{\delta_{\alpha-1}}{\delta_\alpha}\right), \tag{10}$$

where $\delta_\alpha = E_{\alpha+1} - E_\alpha$ is the spacing between the neighboring eigenvalues of the Hamiltonian. The $r$-metric captures short-range correlations among the energy levels. For an integrable system with Poissonian level spacing distribution, the average of $r_\alpha$ over the spectrum gives $r_{\text{Poisson}} \approx 0.39$, while for chaotic systems $r_{\text{GOE}} \approx 0.53$ [53] The latter is the value obtained for full random matrices from Gaussian orthogonal ensembles (GOE) [53].

In Fig. 2 we plot the $r$-metric for four different spin sizes as a function of the number $N$ of excitations in the system. We repeat our calculations for chain lengths

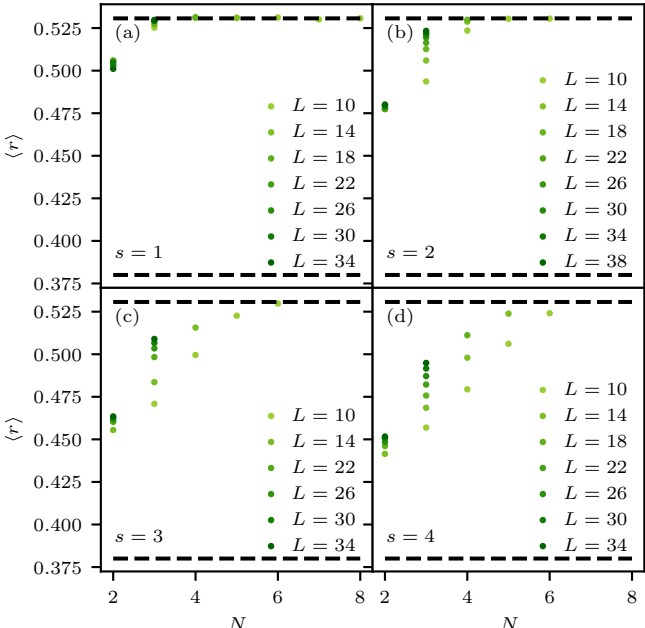

Figure 2. Average of the $r$-metric for four different spin sizes, $s = 1, 2, 3, 4$, as a function of the number of excitations $N$ and various system sizes. Darker circles represent larger chain sizes (see legends). In the calculations of this metric, the top and bottom 15% of the eigenvalues are omitted. The dashed horizontal lines stand for $r_{\text{Poisson}} = 0.39$ and $r_{\text{GOE}} = 0.53$.

ranging from 10 to 38 sites (darker colors indicate larger chains in the figure). Surprisingly, while the classical limit of this model is chaotic, as shown in the previous section, for larger spin sizes, *more* excitations are required to reach signatures of chaos in the quantum domain. The system with spin 1 is fairly chaotic for $N = 3$ and 4, similarly to the spin-1/2 model studied in Refs. [29, 30], but $\langle r \rangle$ for the spin-4 model does not reach $r_{\text{GOE}}$ for the system sizes considered. For a fixed system size and a fixed number of excitations, the $r$-metric in Fig. 2 indicates that a system with a larger spin presents a weaker degree of chaos than its counterpart with a smaller spin.

To better understand why the degree of quantum chaos decreases as the spin is enlarged, we resort to the analysis of the distribution of the off-diagonal elements of the local magnetization, $\hat{S}_z^{L/2}$. For this purpose we take 250 states from the middle of the spectrum and compute the matrix element $S_{z,\alpha\beta}^{L/2} \equiv \langle \alpha | \hat{S}_z^{L/2} | \beta \rangle$ where $\alpha \neq \beta$ are two different eigenstates. Since the variance of the distribution decreases with the Hilbert space dimension $\mathcal{D}$ [48], we normalize the distribution by dividing $S_{z,\alpha\beta}^{L/2}$ by its standard deviation, $\sigma = \text{std}\left(S_{z,\alpha\beta}^{L/2}\right)$.

The resulting distributions for $s = 1, 4$ and $N = 3,4$ can be seen in Fig. 3. The distributions for the spin-1 case in Figs. 3 (a)-(b) are very close to Gaussian (dashed black

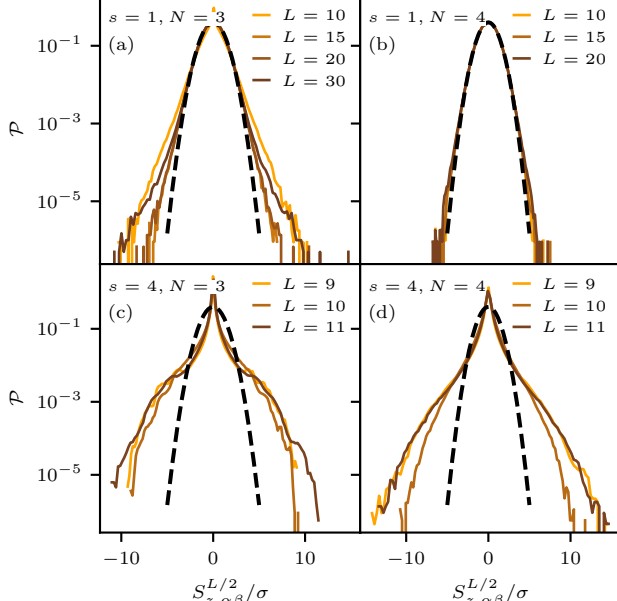

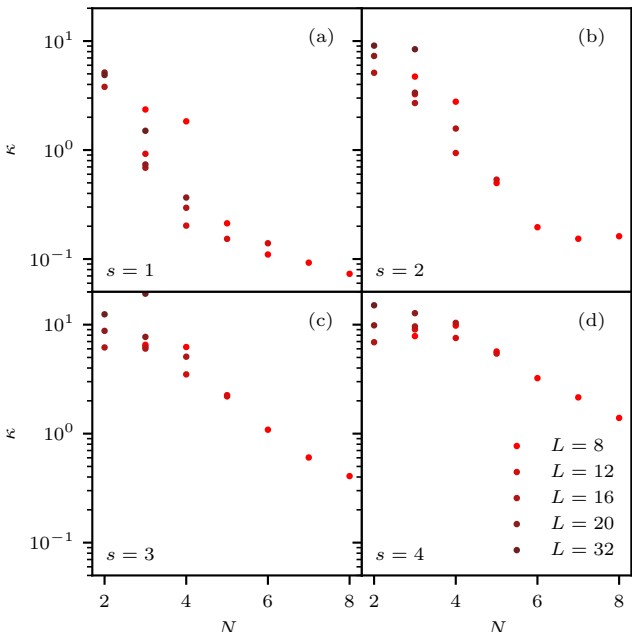

Figure 3. Distributions of the off-diagonal elements of the $\hat{S}_z^{L/2}$ operator for spin 1 (top row) and spin 4 (bottom row) and varying system sizes. Left column corresponds to $N = 3$ excitations and right column to $N = 4$ excitations. We use 250 eigenstates in the middle of the spectrum to compute the off-diagonal elements, and normalize the distributions to have unit variance. The dashed black line shows a Gaussian distribution with unit variance.

Figure 4. Kurtosis of the distributions of the off-diagonal elements of the $\hat{S}_z^{L/2}$ operator for four different spin sizes, $s = 1, 2, 3, 4$, as a function of the number of excitations $N$ and various system sizes. Darker circles represent larger chain sizes (see legends). In the calculations of the kurtosis we used 250 eigenstates from the middle of the spectrum. For a chaotic system the kurtosis is supposed to be zero.

line) for both $N = 3$ [Fig. 3 (a)] and $N = 4$ [Fig. 3 (b)] excitations. However, the spin-4 distributions depicted in in Fig. 3 (c)-(d) are strongly peaked around zero. To quantify how close the distributions are to a Gaussian, we calculate their normalized kurtosis,

$$\kappa = \frac{\langle O_{\alpha\beta}^4 \rangle - \langle O_{\alpha\beta} \rangle^4}{\sigma^4} - 3, \tag{11}$$

which is equal to 0 for a Gaussian distribution. Figure 4 shows the kurtosis of the distributions for $s = 1, 2, 3$ and 4 as a function of the number of excitations $N$ for different system sizes. For $s = 3$ and 4 the dependence of the kurtosis on the number of excitations is non-monotonic around $N = 2, 3$. For $s = 4$, the kurtosis, $\kappa$, only decreases when $N > 4$, and much slower than for $s = 1, 2$. These results confirm that larger spins require more excitations to reach the chaotic regime. Note that the situation does not improve with system size, since for larger $L$, the kurtosis actually moves further away from zero.

Why do larger spins result in a reduced level of quantum chaos? Large-spin chains can contain sites with many excitations, while our disordered XXZ model in Eq. (1) allows only for the motion of one excitation at a time, which makes these clusters of excitations very hard to disassemble. We conjecture that these large and slow moving clusters prevent the onset of strong chaos in our system and propose two mechanisms to reverse this scenario, as discussed next.

## V. ENHANCING QUANTUM CHAOS

In this section we consider two ways to improve the chaotic properties of our finite large-spin system by modifying its Hamiltonian. The first mechanism that we consider is the diversification of the sizes of the clusters of excitations by giving different energies to clusters with different number of excitations. This is achieved with the addition of properly normalized nonlinear magnetization terms to the Hamiltonian,

$$H_1 = H + \frac{\alpha}{s(s+1)} \sum_k \left( \hat{S}_z^k \right)^2, \tag{12}$$

$$+ \frac{\mu}{(s(s+1))^{3/2}} \sum_k \left( \hat{S}_z^k \right)^3,$$

where $\alpha = 0.87, \mu = 0.91$, and similarly to Eq. (1), the normalization of the nonlinear terms is taken to pertain a proper classical limit for $s \to \infty$. The classical limit of this model is

$$H_1^{cl} = H_{cl} + \alpha \sum_k \left( s_z^k \right)^2 + \mu \sum_k \left( s_z^k \right)^3. \tag{13}$$

In Fig. 1, we present the maximal Lyapunov exponents of this model with red points and verify that they closely follow the Lyapunov exponents of the classical disordered XXZ model in Eq. (4). We note that the addition of these terms slightly increases the Lyapunov exponents.

The second quantum chaos enhancement mechanism that we consider is the facilitation of the fragmentation of the clusters of excitations using nonlinear ladder operators,

$$H_2 = H_1 + \tag{14}$$
$$+ \sum_{k=1}^{L-1} \sum_{n=2}^{s} \frac{J_{xy}}{s^n (s+1)^n} \left[ \left( \hat{S}_+^k \hat{S}_-^{k+1} \right)^n + \left( \hat{S}_+^{k+1} \hat{S}_-^k \right)^n \right].$$

The added terms move $k-$excitations between neighboring sites with $2 \leq k \leq s$. While the classical limit of this model is well defined, the derivation of its Hamiltonian in a closed form is cumbersome, because the sum over $k$ of the nonlinear ladder operators has to be computed explicitly, so we do not show it here.

The results for the Hamiltonians in Eq. (12) and Eq. (14) show a systematic improvement for all considered quantum chaos metrics when compared to the disordered XXZ model in Eq. (1). Figure 5 shows the $r$-metric of all three models computed for spin 1 [Figs. 5(a)-(b)] and spin 4 Figs. 5(c)-(d)] as a function of the Hilbert space dimension, $\mathcal{D}$. The spin-1 case is already fairly chaotic for the Hamiltonian (1), so the addition of the new terms in Eq. (12) and Eq. (14) do not affect the values of $\langle r \rangle$. However, for spin 4, adding the nonlinear magnetization terms in Eq. (12) dramatically improves the degree of quantum chaos for both $N = 3$ and 4. The addition of the nonlinear ladder operators in Eq. (14) is even more effective and leads to strong level repulsion for as few as 3 excitations, making the results analogous to the case of spin-1/2 studied in Ref. [30]. This significant enhancement of the degree of quantum chaos is corroborated by the other chaotic metrics considered in this work, namely the distributions of off-diagonal observables and the kurtosis of these distributions (not shown).

## VI. DISCUSSION

The driving question of this work is whether increasing the spin size of a one-dimensional spin model can reduce the number of spin excitations needed to achieve quantum chaos. For this purpose we studied the large spin limit of a disordered XXZ chain. For a single spin-1/2 excitation this model is integrable and localized via the Anderson localization mechanism, while at zero $z$-magnetization it is chaotic. In previous studies of spin-1/2 chains, it was established that at least 3 spin excitations are required to achieve quantum chaos. By considering chaotic metrics based on both the eigenvalues

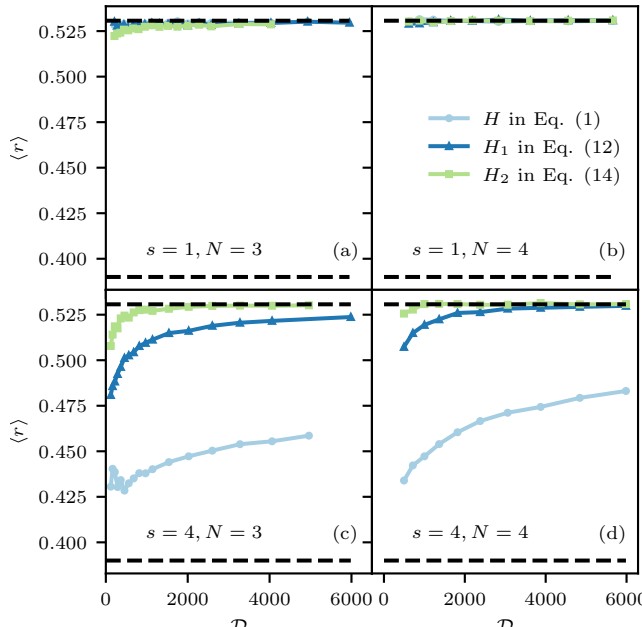

Figure 5. Averaged $r$-metric as a function of the Hilbert space dimension $\mathcal{D}$ for spin-1 models(top row) and spin-4 models (bottom row). The left column shows $N = 3$ excitations and the right column shows $N = 4$ excitations. Circles correspond to the disordered XXZ model (1), triangles to model (12), and squares to model (14). The dashed horizontal lines stand for $r_{\text{Poisson}} = 0.39$ and $r_{\text{GOE}} = 0.5307$.

and the eigenstates, we found that, although the classical limit of the disordered XXZ chain is chaotic for very low magnetization, the large spin version of the quantum XXZ model, surprisingly, shows significantly reduced chaotic behavior, compared to its spin-1/2 counterpart. We attribute this phenomenon to the occurrence of clusters of excitations concentrated on one or a number of neighboring sites, which makes the clusters hard to decompose and the corresponding dynamics slow. We proposed two mechanisms to enhance quantum chaotic behavior: by the introduction of additional relaxation channels for the clusters in the face of a collection of ladder operators and by adding nonlinear onsite magnetization. We numerically verified that the introduction of these terms allows to achieve quantum chaos for $N = 3$ excitations, similarly to the spin-1/2 systems. Interestingly, the introduction of additional relaxation channels enhances *classical* chaos only slightly. Our study suggests that care should be taken in considering large-spin limits of quantum models, since certain relaxation channels become inherently slow. It also suggests that classical chaos and quantum chaos might not be so tightly bound.

This research was supported by a grant from the United States-Israel Binational Foundation (BSF, Grant No. 2019644), Jerusalem, Israel, and the United

States National Science Foundation (NSF, Grant No. DMR−1936006), and by the Israel Science Foundation (grants No. 527/19 and 218/19).

————————

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
