# Peer review of "Chaos enhancement in large-spin chains"

_SciPost Physics_

## Round 1 · Referee Report · Anonymous (Referee 1) · 2022-5-12

Report

In their work, the authors address the chaotic features of spin chains described by a quantum non-integrable Hamiltonian, the anisotropic Heisenberg model in the presence of a weak disordered field and its connection to the -expected- classical limit; in which the magnitude of the local spin is very large. In particular, the authors are interested in approximating the minimum number of excitations required to observe the signatures of chaos both in the low-magnitude and large-magnitude spin cases.
It is shown that the classical model (for which the spin magnitude is large and local spin matrices commute) is chaotic, in the sense of the presence of a positive Lyapunov exponent that signals this regime.
The authors then proceed to evaluate numerical quantities that are considered to signal chaotic features, such as level spacing statistics and probabilistic distributions of off-diagonal matrix elements of local observables in the energy eigenbasis. Their results indicate that increasing the magnitude of the local spin reduce chaotic behaviour, so they propose to introduce certain perturbations that might enhance chaos in this regime.
I find the paper easy to read and I particularly like their conclusion. I, however, would not recommend publication unless the following points are appropriately addressed:

Requested changes

1- Section IV: Random matrices fail to describe the richness of the structure of microscopic Hamiltonians, such as the Heisenberg model presented here, as posited by the eigenstate thermalisation hypothesis. Not only are treatments of random matrices oblivious to energetic considerations (effectively only partially describing infinite-temperature ensembles), but it has now been pointed out that the matrix elements of local observables in the energy eigenbasis cannot be independent and identically distributed random variables. Granted, they appear to be so in very small frequency scales. This was suggested to be reason behind the exponential growth of out-of-time order correlators (and positive Lyapunov exponents) by Foini and Kurchan in: https://journals.aps.org/pre/pdf/10.1103/PhysRevE.99.042139 The frequency-dependent presence of statistical correlations was later characterised by Richter et. al. in: https://journals.aps.org/pre/abstract/10.1103/PhysRevE.102.042127 Furthermore, it was later shown that the frequency-dependent presence of statistical correlations results in an important consequence in the dynamics of high order correlation functions by Brenes et. al. in: https://journals.aps.org/pre/abstract/10.1103/PhysRevE.104.034120 I believe the authors need to discuss these considerations in further detail, as I find the first paragraph in Section IV to be an oversimplification of what has now been described as intricate qualities of quantum chaos. 2- The authors employ level spacing statistics and distributions of matrix elements of local observables in the energy eigenbasis to infer chaotic behaviour. What are the caveats of considering these quantities as sole indicators? Level spacing distributions, as presented in this paper for instance, only considers local effects between the eigenvalues. Quantum chaos has been characterised from the perspective of deformations in the adiabatic gauge potential: https://journals.aps.org/prx/abstract/10.1103/PhysRevX.10.041017 and spectral form factors: https://journals.aps.org/pre/abstract/10.1103/PhysRevE.102.062144 How do these connect to each other? Are we allowed to assume that a generic system with a GOE or GUE distribution of spacings will in turn display all the qualities of a chaotic system, such as, for instance, the emergence of hydrodynamical behaviour of transport? I believe a discussion on these topics is of the essence. 3- Section VI: "...In previous studies of spin- 1/2 chains, it was established that at least 3 spin excitations are required to achieve quantum chaos..." This statement needs citations. The authors are probably referring to their previous submission to SciPost. 4- Section III: The algorithms and methods described in Refs. [34] and [35] should, at least, be described in a minimal fashion in the paper. 5- The results exposed in Figure 1 need to be further described. I, for instance, do not understand why there are several points of MLE for a single phase deviation $\delta \theta$. At first I thought each point for a given $\delta \theta$ corresponded to a disorder realisation, however, in the caption it is stated that each point is an ensemble average. I believe further clarification is required.

I believe these are additions that would be appreciated by a future reader.

  • validity: good
  • significance: good
  • originality: ok
  • clarity: high
  • formatting: excellent
  • grammar: perfect

Author:  Yevgeny Bar Lev  on 2023-03-30  [id 3523]

(in reply to Report 1 on 2022-05-12)

The referee writes:

1- Section IV: Random matrices fail to describe the richness of the structure of microscopic Hamiltonians, such as the Heisenberg model presented here, as posited by the eigenstate thermalisation hypothesis. Not only are treatments of random matrices oblivious to energetic considerations (effectively only partially describing infinite-temperature ensembles), but it has now been pointed out that the matrix elements of local observables in the energy eigenbasis cannot be independent and identically distributed random variables. Granted, they appear to be so in very small frequency scales. This was suggested to be reason behind the exponential growth of out-of-time order correlators (and positive Lyapunov exponents) by Foini and Kurchan in: https://journals.aps.org/pre/pdf/10.1103/PhysRevE.99.042139 The frequency-dependent presence of statistical correlations was later characterised by Richter et. al. in: https://journals.aps.org/pre/abstract/10.1103/PhysRevE.102.042127 Furthermore, it was later shown that the frequency-dependent presence of statistical correlations results in an important consequence in the dynamics of high order correlation functions by Brenes et. al. in: https://journals.aps.org/pre/abstract/10.1103/PhysRevE.104.034120 I believe the authors need to discuss these considerations in further detail, as I find the first paragraph in Section IV to be an oversimplification of what has now been described as intricate qualities of quantum chaos.

The Referee is totally right. Full random matrices “fail to describe the richness of structure of microscopic Hamiltonians”. This is exactly why Wigner introduced, back in the 50’s, the Wigner band random matrices and this is also why the nuclear physics and quantum chaos communities have been using two-body-random ensembles (TBRE) for decades. In any realistic model, there are correlations among the elements of the Hamiltonian matrix that affect the structure of the eigenstates and “the matrix elements of local observables”. In the beginning of Sec.IV, we had used words such as “close”, “almost”, “away from the edges” to distinguish realistic chaotic models from full random matrices. We now added a paragraph to further stress the difference. Notice, however, that we are not using full random matrices in our work. We simply verify how much the properties of our system depart from full random matrix behaviors as we increase spin size, which is a rather counterintuitive result.

The referee writes:

2- The authors employ level spacing statistics and distributions of matrix elements of local observables in the energy eigenbasis to infer chaotic behaviour. What are the caveats of considering these quantities as sole indicators? Level spacing distributions, as presented in this paper for instance, only considers local effects between the eigenvalues. Quantum chaos has been characterised from the perspective of deformations in the adiabatic gauge potential: https://journals.aps.org/prx/abstract/10.1103/PhysRevX.10.041017 and spectral form factors: https://journals.aps.org/pre/abstract/10.1103/PhysRevE.102.062144 How do these connect to each other? Are we allowed to assume that a generic system with a GOE or GUE distribution of spacings will in turn display all the qualities of a chaotic system, such as, for instance, the emergence of hydrodynamical behaviour of transport? I believe a discussion on these topics is of the essence.

As explained above, the interest of this work is not in the proximity to random matrix behavior, but what takes our system away from it. If the r-statistics indicates lack of chaos, then the analysis of spectral correlations via short-range (r-statistic, level spacing distribution) or long-range (rigidity, level number variance, spectral form factor) correlations will all indicate lack of chaos. Increasing the spin size takes our model away from chaos, that’s the subject of our work. It is an interesting question whether “a generic system with a GOE or GUE distribution of spacings will in turn display all the qualities of a chaotic system”, but answering this question is not the goal of our work.

The referee writes:

3- Section VI: "...In previous studies of spin- 1/2 chains, it was established that at least 3 spin excitations are required to achieve quantum chaos..." This statement needs citations. The authors are probably referring to their previous submission to SciPost.

Thanks. We have added the appropriate references.

The referee writes:

4- Section III: The algorithms and methods described in Refs. [34] and [35] should, at least, be described in a minimal fashion in the paper.

We have added a short description of the algorithm [34]. We chose not to describe the algorithm in [35], since it is a general algorithm for the solution of stiff differential equations.

The referee writes:

5- The results exposed in Figure 1 need to be further described. I, for instance, do not understand why there are several points of MLE for a single phase deviation $\delta\theta$. At first I thought each point for a given $\delta\theta$ corresponded to a disorder realisation, however, in the caption it is stated that each point is an ensemble average. I believe further clarification is required.

In Fig.1, each point for a given $ \theta/\pi $ represents a maximal Lyapunov exponent computed for a different realization of disorder and initial xy-angles of the rotors. We have adjusted the caption correspondingly, and added clarification in the text.

---

## Round 1 · Referee Report · Anonymous (Referee 2) · 2022-7-28

Strengths

1- Identifies instance of model with classical chaos but with slow approach to quantum chaos especially for large spins where naively quantum chaos might be expected to be more pronounced. 2- The ergodic properties of model studied in the paper are of general interest especially in the spin one-half case. This paper partially addresses the physics of the higher spin case.

Weaknesses

Possible weaknesses:

1- The authors could strengthen their introduction with a discussion of the broader significance of their findings. 2- The non-gaussian distributions and non-GOE r-statistics are interesting results but there appears to be little understanding of these features. The discussion of slowly relaxing clusters sounds reasonable but various aspects of this picture have not been clearly established it seems. 3- Although the paper presents results for various system sizes, spin sizes and magnetization sectors (and despite figures showing some trends) there is no systematic overview of where anomalous distributions can be observed. 4- The results are fairly clear on the finite size features of r-statistics and off-diagonal matrix element distributions. But the paper seems to miss an opportunity to explain in the text the significance for the thermalizing behavior in the thermodynamic limit (to the degree this can be accessed by finite size scaling and despite the fact that various system sizes were examined) and the connection to the spin one-half case - for example by tuning the disorder strength.

Report

The general context for the work is how higher spin models approach the classical limit and the degree to which s>1/2 are like their spin-1/2 counterparts. The particular higher spin models studied here are of interest also from the perspective of MBL.
Perhaps the most appropriate acceptance criterion here is #3: "Open a new pathway in an existing or a new research direction, with clear potential for multipronged follow-up work".

At the moment the paper has some tantalizing results that could be presented in a more systematic way to help guide future work in these directions.

Requested changes

  • The matrix element distributions suggest persistent non-gaussianity for the system sizes investigated. Is this consistent with the finite size scaling of the r-statistics? i.e. it would be good to see the r-statistics data re-plotted versus L.
  • The data plotted in Fig 3 is the most fine-grained picture of the anomalous large spin behavior. What is the regime where the distribution is gaussian and what is the regime where it is cusped as S, N and disorder strength are varied? Can one conclude anything from these trends? Maybe a "phase diagram" would be helpful here.
  • Is there some understanding of the cusps at zero in Fig 3 (c,d)?
  • Relatedly, the authors conjecture that clusters of excitations can have slow dynamics but the results that are presented are r-stats and matrix element distributions. Is it clear that the fat tails are coming from such clusters in real space?
  • Is it clear that the departure from GOE statistics can be accounted for by states contributing to the fat tails in the matrix element distribution?
  • I do not understand the claimed "slow-relaxing clusters". This is despite the plausible statements just before section V and section V itself where additional terms are added to the Hamiltonian that assist the system towards chaos and that target highly excited local states. My confusion relates to (i) the discrepancy between static observables and dynamical statements in the paper (ii) the role of disorder. Concretely, it would be good to present some measure (e.g. the entanglement) across the spectrum to look for the distribution of anomalous states and their density. How does this picture vary with disorder strength (in the relatively low disorder regime)?
  • Again on the slow-relaxing clusters. It would be good to have a concrete picture either directly tied to calculations of the dynamics or features of the spectrum. e.g. Is there any sense in which they are emergent single particle states (e.g. bound states) that localize in the disorder potential and bias the spectrum towards Poissonian r-stats? If not, how should one think of them and their effects on the spectrum and especially the fat tails and cusps in the matrix element distribution.

  • validity: high
  • significance: ok
  • originality: good
  • clarity: high
  • formatting: excellent
  • grammar: excellent

Author:  Yevgeny Bar Lev  on 2023-03-30  [id 3522]

(in reply to Report 2 on 2022-07-28)

The referee writes:

The matrix element distributions suggest persistent non-gaussianity for the system sizes investigated. Is this consistent with the finite size scaling of the r-statistics? i.e. it would be good to see the r-statistics data re-plotted versus L.

Our response:

Figure 2 has the r-statistics data plotted as a function of the number of excitations N for several system sizes L. For each N, the points get darker as the system size increases. Such a presentation allows us to see the trend for all studied Ns. Dependence of the r-statistics on D (the Hilbert space dimension, and therefore L), is plotted in Fig 5. Taking s=4 and N=3, for example, which corresponds to highly non-Gaussian distributions [Fig 3(c)], the r-statistics [Fig.2(d)] for the largest system at hand is far below its GOE value. Therefore the results for both metrics are consistent.

The referee writes:

The data plotted in Fig 3 is the most fine-grained picture of the anomalous large spin behavior. What is the regime where the distribution is gaussian and what is the regime where it is cusped as S, N and disorder strength are varied? Can one conclude anything from these trends? Maybe a "phase diagram" would be helpful here.

Our response: Figure 3 analyzes the distribution for various S and N values. We would like to stress that the role of disorder in our work is limited to breaking any residual symmetries. This is why we take it to be fairly weak compared to other parameters (W=0.5). For larger W, localization effects will kick-in. Since the goal of the work was to study chaoticity in large spin models, we deliberately stayed away from large disorder strengths.

The referee writes:

Is there some understanding of the cusps at zero in Fig 3 (c,d)?

Our response: The cusps correspond to prevalence of zeros in the matrix of the local operators, which means that the matrix is effectively sparse. This is normally seen as a feature of integrability. When the system is close to an obvious integrable limit, such as W=0 and S=1/2, the cusp can be motivated from a perturbation theory. Here we don’t have an obvious small or large parameter.

The referee writes:

Relatedly, the authors conjecture that clusters of excitations can have slow dynamics but the results that are presented are r-stats and matrix element distributions. Is it clear that the fat tails are coming from such clusters in real space?

Our response: The existence of clusters is an attempt to reason why the Hamiltonian (14) is much more chaotic, compared to (1). Following the suggestions in the referees’ reports, we have tried to obtain direct evidence of the existence of clusters, or alternatively, an existence of special eigenstates. We couldn’t clearly identify special eigenstates, both in terms of entanglement entropy, as also the composition of the eigenstates (their overlap with “cluster” states). Looking at the dynamics, starting from clustered excitation, also doesn’t appear to show significantly slower relaxation between Hamiltonian (14) and Hamiltonian (1). As a result of this investigation, we agree with the Referee, that there is no hard evidence of the cluster states being the reason for the apparent lack of chaoticity at larger spins. We have rewritten the manuscript correspondingly.

In the process, we have examined in more detail the large spin limit (s>20), since we suspected that the difference between Hamiltonian (14) and (1) might more clearly establish itself in this limit. Interestingly, while in the s->inf limit the off-diagonal elements of both Hamiltonians become negligible, Hamiltonian (14) is still chaotic, even for very large spins. We have added this discussion to the text as a separate section.

The referee writes:

Is it clear that the departure from GOE statistics can be accounted for by states contributing to the fat tails in the matrix element distribution?

Our response: No, it is not clear. It is an interesting question to consider, how the r-statistics and distribution of matrix elements are related in weakly chaotic many-body quantum systems, though this was not the goal of our work... We notice, however, that there are significant differences between systems with few and many excitations. In 3D chaotic systems with one excitation, for example, where the GOE statistics is evident, but the density of states is not Gaussian, the structure of the eigenstates is also not Gaussian.

The referee writes:

I do not understand the claimed "slow-relaxing clusters". This is despite the plausible statements just before section V and section V itself where additional terms are added to the Hamiltonian that assist the system towards chaos and that target highly excited local states. My confusion relates to (i) the discrepancy between static observables and dynamical statements in the paper (ii) the role of disorder. Concretely, it would be good to present some measure (e.g. the entanglement) across the spectrum to look for the distribution of anomalous states and their density. How does this picture vary with disorder strength (in the relatively low disorder regime)? - Again on the slow-relaxing clusters. It would be good to have a concrete picture either directly tied to calculations of the dynamics or features of the spectrum. e.g. Is there any sense in which they are emergent single particle states (e.g. bound states) that localize in the disorder potential and bias the spectrum towards Poissonian r-stats? If not, how should one think of them and their effects on the spectrum and especially the fat tails and cusps in the matrix element distribution.

Our response: We have addressed these points in the answers above.

---

## Editorial Decision

resubmitted